# Impact of Smoking Cessation and Charlson Comorbidity Index on Influenza Vaccination Efficacy in COPD Patients

**DOI:** 10.3390/microorganisms12071437

**Published:** 2024-07-16

**Authors:** Hui-Chuan Chang, Shih-Feng Liu

**Affiliations:** 1Department of Respiratory Therapy, Kaohsiung Chang Gung Memorial Hospital, Kaohsiung 833, Taiwan; elaine11142@cgmh.org.tw; 2Division of Pulmonary and Critical Care Medicine, Department of Internal Medicine, Kaohsiung Chang Gung Memorial Hospital, #123, Ta-Pei Road, Niaosong District, Kaohsiung 833, Taiwan; 3College of Medicine, Chang Gung University, Taoyuan 333, Taiwan

**Keywords:** chronic obstructive pulmonary disease, influenza vaccination, Charlson Comorbidity Index, smoking cessation

## Abstract

Chronic obstructive pulmonary disease (COPD) patients are particularly susceptible to respiratory infections like influenza, which exacerbate symptoms and increase healthcare utilization. While smoking cessation and influenza vaccination are recommended preventive measures, their combined impact on healthcare resource utilization is underexplored. The Charlson Comorbidity Index (CCI) assesses comorbidity burden in COPD patients and may influence healthcare outcomes. We conducted a retrospective analysis of 357 COPD patients, evaluating smoking cessation success over one year and influenza vaccination receipt, stratifying patients by CCI scores. Healthcare utilization outcomes included emergency room visits, hospitalizations, and medical expenses. Results showed that 51.82% of patients quit smoking and 59.66% received influenza vaccination, with higher comorbidity prevalence in advanced COPD stages (*p* = 0.002). Both smoking cessation and influenza vaccination independently correlated with decreased emergency room visits, hospital admissions, days, and costs. Patients who both quit smoking and received influenza vaccination exhibited the lowest healthcare utilization rates. In conclusion, smoking cessation and influenza vaccination significantly reduce healthcare resource utilization in COPD patients, with the combination yielding synergistic benefits, particularly in those with lower CCI scores. Integrating these interventions and comorbidity management in COPD strategies is essential for optimizing patient outcomes and healthcare efficiency.

## 1. Introduction

Chronic obstructive pulmonary disease (COPD) is a complex respiratory disorder marked by enduring airflow limitation, typically coupled with chronic bronchitis and emphysema [1]. It plays a substantial role in global health, contributing significantly to worldwide morbidity and mortality rates [2]. COPD patients grapple with a range of symptoms, including persistent coughing, wheezing, breathlessness, and diminished exercise tolerance, all of which severely impact their quality of life [3].

COPD patients’ amplified susceptibility to respiratory infections like influenza is of particular concern [4]. Such infections can incite COPD exacerbations characterized by exacerbating symptoms and deteriorating lung function, culminating in hospital admissions and potentially even death [5]. Influenza poses a particular threat given its capacity to cause severe respiratory illness and complications, further undermining respiratory function in an already susceptible population [6].

Considering the increased risk of influenza-related morbidity and mortality among COPD patients, influenza immunization is heavily advocated as a preventive measure [7]. Influenza vaccination has effectively reduced the incidence of influenza infection, hospitalizations, and mortality among COPD patients [8]. However, the efficiency of influenza vaccination in this group may be moderated by various factors, including personal attributes and coexisting health conditions.

One such influential factor is smoking cessation, which is pivotal in managing COPD [9]. Smoking is the primary origin of COPD, and continuous smoking exacerbates disease progression and respiratory symptoms [10]. Ceasing smoking not only decelerates the decline in lung function but also lessens the likelihood of exacerbations and enhances overall health outcomes in COPD patients [11]. Nonetheless, the impact of smoking cessation on the efficacy of influenza vaccination is a topic of ongoing study.

Comorbidities that frequently coexist with COPD may also affect the outcomes of vaccination [12]. The Charlson Comorbidity Index (CCI) is a recognized tool for evaluating the burden of comorbidity, covering conditions like cardiovascular disease, diabetes, and renal dysfunction, among others [13]. These comorbidities may influence immune responses to vaccination, potentially predisposing COPD patients to a higher likelihood of influenza-related complications.

Grasping the interactions between smoking cessation, comorbidities as captured by the CCI, and the effectiveness of influenza vaccination is essential for enhancing preventive strategies in COPD patients. Personalized vaccination strategies based on individual patient profiles and addressing modifiable risk factors like smoking and comorbidities could increase vaccine effectiveness and better health outcomes in this vulnerable group [14]. Therefore, additional research is needed to clarify these relationships and guide evidence-based guidelines for influenza vaccination in individuals with COPD.

## 2. Method

This study conducted a retrospective analysis to explore the individual analysis of the Charlson Comorbidity Index for smoking cessation in patients with chronic obstructive pulmonary disease (COPD) after receiving influenza vaccination. Patient data were collected from the database of Kaohsiung Chang Gung Medical Center, which ranks first in the application for medical service points in the Nanping area. From January to October 2018, patients diagnosed with COPD in outpatient clinics were selected based on the main diagnosis of ICD-10 J44 appearing five times or more. According to a study published in 2017 in an international journal, early COPD patients were observed continuously for six years, with an average of 4.5 outpatient visits per year. In 2012, the average number of outpatient visits was 4.9 ± 5.2 times [5]. Considering that COPD patients typically require bronchodilator drugs for approximately two months and regular follow-up visits to outpatient clinics to monitor their condition, the subjects of this study were selected based on the main diagnosis of ICD-10 J44 appearing five times or more in outpatient diagnosis records. The impact of smoking cessation for one year on the hospitalization days, emergency room visits, medical expenses, and occurrences of respiratory failure in COPD patients after receiving influenza vaccination was observed due to diagnoses of acute respiratory tract infections, pneumonia, and influenza.

### 2.1. Charlson Comorbidity Index (CCI)

The Charlson Comorbidity Index (CCI) is often used to assess the impact of comorbid conditions on COPD patients, providing a comprehensive understanding of their overall health status. Subjects were further stratified into three groups based on their Charlson Comorbidity Index (CCI) scores [7], which serve to adjust for comorbidity status. A higher CCI score indicates a greater burden of comorbid health conditions. When analyzing the distribution of subjects across different CCI scores, the highest score observed is 10. Upon analysis, the median score is 5, and we used a standard deviation of 1.3 as the cutoff point. We divided into three groups (i.e., CCI I, CCI scores: 0–3; CCI II, CCI scores: 4–6; CCI III, CCI scores: 7 or higher).

### 2.2. Influenza Vaccine

Taiwan provided only trivalent influenza vaccine (TIV) through the public health system. The trivalent influenza vaccine protects against two strains of influenza A and one strain of influenza B. Ensure the efficacy of the influenza vaccine and good work in preventing influenza infection. In this study, we define the vaccinated individuals as those who have received the influenza vaccine within the past six months.

### 2.3. Smoking Cessation

Individuals who had successful smoking cessation abstained from smoking for at least one year. Participants were classified based on their smoking cessation status: individuals who successfully ceased smoking for at least one year were categorized as smoking cessation: YES, while those who did not successfully quit smoking were considered smoking cessation: NO, as confirmed by chart records.

### 2.4. Statistical Analysis

Descriptive statistics were employed to analyze various study variables related to the Charlson Comorbidity Index (CCI), across three major groups. The distribution of observed characteristics such as gender, age, smoking cessation, influenza vaccination status, emergency room visits, hospitalization days, and medical expenses was examined within each group.

Inferential statistics included chi-square tests for categorical variables and one-way ANOVA for independent samples were applied to continuous variables to compare among the three CCI groups, followed by the Bonferroni post hoc test. The independent variable in this study was the receipt of influenza vaccination. A multiple linear regression analysis was conducted to explore the relationship between emergency department visits, hospitalization rates, and hospital length of stay with variables, including age, sex, smoking cessation, influenza vaccination status, and the Charlson Comorbidity Index. Furthermore, logistic regression analysis was used to assess the impact across CCI three groups on the frequency of emergency and hospital utilization. Data analysis was performed using STATA Version 14 (College Station, TX 77845, USA). Statistical significance was defined as a two-sided *p*-value of less than 0.05.

## 3. Results

Table 1 illustrates the distribution of the study sample, which comprised 357 patients. The patients were categorized into three groups: CCI I, CCI II, and CCI III, with each group consisting of 90, 90, and 177 patients, respectively. The average age of participants was 71.69 ± 8.4 years in CCI I, 71.03 ± 10.04 years in CCI II, and 70.57 ± 9.63 years in CCI III. Males comprised 92.44% of the sample, while females accounted for 7.56%, totaling 27 individuals. Out of the total study participants, 213 (59.66%) had received an influenza vaccination. Specifically, the vaccination rates in CCI I, CCI II, and CCI III were 49 (54.4%), 52 (57.78%), and 112 (63.28%) patients, respectively. Additionally, as shown in Figure 1, participants were identified with a higher potential for neither undergoing smoking cessation nor receiving an influenza vaccination, with this higher proportion seen in the CCI III group. Conversely, the lowest proportion of participants who successfully quit smoking and received an influenza vaccination was observed in the CCI I group.

Table 2 shows that after multiple regression analyzing emergency visits, the coefficient for age was −0.02 (*p* = 0.06), and the coefficient for gender was 0.29 (*p* = 0.61). However, the coefficients for influenza vaccination and smoking cessation were significant, at −0.86 (*p* = 0.00 **) and −1.25 (*p* = 0.00 **), respectively. Using CCI I as the baseline, the coefficient for CCI II compared to CCI I was 1.24 (*p* = 0.00 **), and the coefficient for CCI III compared to CCI I was 1.76 (*p* = 0.00 **). For hospitalization frequencies, age and gender showed no significant relation, with p-values of 0.06 and 0.07, respectively. However, there were significant relations between influenza vaccination, smoking cessation, and CCI groups. The coefficient for influenza vaccination was −0.59 (*p* = 0.00 **), and for smoking cessation, it was −0.69 (*p* = 0.00 **). Compared to CCI I, the coefficient for CCI II was 0.54 (*p* = 0.00 **), and for CCI III, it was 0.90 (*p* = 0.00 **). There were no significant relations between hospital length of stay and age (*p* = 0.9) or gender (*p* = 0.06). However, significant relations were observed with influenza vaccination, smoking cessation, and CCI groups. The coefficient for influenza vaccination was −5.57 (*p* = 0.00 **), and for smoking cessation, it was −3.15 (*p* = 0.00 **). Compared to CCI I, the coefficient for CCI II was 2.28 (*p* = 0.193), and for CCI III, it was 4.91 (*p* = 0.002 *). Notably, only CCI III showed a significant difference in hospital length compared to CCI I, whereas there was no significant difference between CCI II and CCI I.

In Table 3, the analysis of medical utilization was conducted based on whether the influenza vaccine was administered and whether smoking cessation was successful. The results showed that for CCI II and CCI III, receiving the influenza vaccine or successful smoking cessation led to a significant (p:00**) decrease in the coefficients for emergency visits, hospitalization frequencies, and hospital length of stay. In contrast, for CCI I, only hospital length of stay was significantly associated with the administration of the influenza vaccine and the success of smoking cessation.

Figure 2 shows that in the CCI I group, there is no significant association between emergency utilization and hospital utilization, regardless of whether the influenza vaccine was administered or whether smoking cessation was successful. However, the odds ratio for those who received the influenza vaccine compared to those who did not, those who successfully quit smoking compared to those who did not, and those who had both conditions all show a lower probability of emergency utilization and hospital utilization.

In the CCI II group, the odds ratio for emergency utilization is significantly lower among those who received the influenza vaccine compared to those who did not. When considering both the administration of the influenza vaccine and the success of smoking cessation, the odds ratios for both emergency utilization and hospital utilization are less than 1 for those who received the vaccine and successfully quit smoking.

In the CCI III group, there is a significant association between emergency utilization and hospital utilization, as well as a significant association between the administration of the influenza vaccine and the success of smoking cessation. The odds ratios for emergency utilization and hospital utilization are less than 1 for those who received the influenza vaccine and successfully quit smoking.

Figure 3 shows the odds ratios analysis of emergency utilization and hospital utilization for individuals who received the influenza vaccine and successfully quit smoking in CCI I, CCI II, and CCI III. The analysis indicates that under the same conditions of receiving the influenza vaccine and successful smoking cessation, the odds ratios for emergency utilization and hospital utilization are all less than 1 across CCI I, CCI II, and CCI III. However, CCI III shows a higher likelihood of emergency utilization and hospital utilization compared to CCI II and CCI I. There is no significant difference between CCI III and CCI II in terms of emergency utilization, but there is a significant difference in hospital utilization. When comparing CCI II with CCI I and CCI III with CCI I, there are significant differences in emergency utilization and hospital utilization.

## 4. Discussion

Our study’s findings emphatically highlight the significance of smoking cessation, influenza vaccination, and the Charlson Comorbidity Index (CCI) in lessening COPD’s burden and refining the usage of healthcare resources among affected patients. In analyzing these interventions’ implications on COPD outcomes and patterns of healthcare utility, we gleaned several decisive insights.

Primarily, we discerned a prominent relationship between smoking cessation and a decline in healthcare resources among COPD patients. Individuals who successfully relinquished smoking exhibited fewer emergency department consultations, hospital admissions, and related costs compared to persistent smokers [11,15]. This observation harmonizes with current evidence proposing that smoking cessation not only decelerates the advance of COPD but also curtails the regularity and severity of exacerbations, consequently reducing healthcare dependency.

Secondly, influenza vaccination emerged as a safeguard against unfavorable COPD outcomes and reduced healthcare resource consumption [16]. Patients immunized against influenza had lower rates of emergency department visits, hospital admissions, and related costs compared with their unvaccinated counterparts [5]. These data accentuate vaccination’s necessity in averting influenza-related exacerbations and complications in COPD patients, hence alleviating the strain on healthcare provision.

Our analysis additionally examined the impact of comorbidities, as reflected by the Charlson Comorbidity Index (CCI), on COPD outcomes and healthcare consumption [17]. We discovered that the elevated CCI scores, signaling a greater burden of comorbidity, were linked with intensified use of healthcare resources across all examined groups. Notwithstanding, even when comorbidities were present, smoking cessation and influenza vaccination remained effective in diminishing healthcare reliance among COPD patients.

By incorporating CCI in our evaluation, we have gleaned invaluable insights into the effects of comorbidities on managing COPD and using healthcare resources. COPD patients with higher CCI scores may present more complex healthcare needs and increased susceptibility to adverse results, underscoring the requirement for customized interventions to refine their care [18]. Despite the hurdles imposed by comorbidities, our observations imply that both smoking cessation and influenza vaccination continue to confer substantial advantages in limiting healthcare consumption in this group.

Crucially, our research revealed reciprocated benefits when smoking cessation, influenza immunization, and the management of comorbidities were amalgamated into COPD care. COPD patients who quit smoking and received influenza vaccinations had lower CCI scores and registered the least frequented emergency department visits and hospitalizations. This finding underscores adopting a comprehensive approach to managing COPD, tackling modifiable risk factors, precluding exacerbations, and efficiently managing comorbidities.

These findings align with the previous studies that have shown the benefits of smoking cessation and vaccination in improving COPD outcomes. For example, a systematic review revealed that influenza vaccination significantly reduced the incidence of exacerbations, hospitalizations, and outpatients in COPD patients [19]. Influenza vaccination has demonstrated a certain degree of effectiveness in reducing laboratory-confirmed influenza-associated hospitalizations in COPD patients. However, to further optimize its impact, improvement in the vaccine and the implementation of other preventive strategies are essential [20]. Additional preventive measures, such as smoking cessation, can also be highly beneficial. Patients who stop smoking can experience immediate benefits from both interventions, leading to improved lung function and respiratory symptoms, thereby enhancing their quality of life [21].

### Limitations

Firstly, the study’s design does not allow for the determination of causal relationships. Additional clinical outcome research is necessary to confirm the causality of the observed effects. Secondly, despite initially considering a large number of cases, it was necessary to exclude many patients to ensure data integrity and properly account for factors such as comorbidities, the specific timing of comorbidity diagnosis or treatment, smoking cessation status, and influenza vaccination. The relatively small number of included patients raises questions about the generalizability of the findings to the broader COPD population. Thirdly, the study’s database was restricted to one medical center, which limits the applicability of the results to other COPD patient groups. Differences in healthcare settings and patient demographics across various centers could affect the outcomes. Fourthly, the healthcare utilization of COPD patients is influenced by more than just smoking cessation and influenza vaccination. Other factors, such as socioeconomic status, ethnicity, the extent of current and previous smoking, and comorbidities, might also affect the observed results, but these were not thoroughly addressed in this study.

## 5. Conclusions

To conclude, our study underlines the profound influence of smoking cessation, influenza vaccination, and the management of comorbidities in decreasing the reliance on healthcare resources in COPD patients. These interventions collaboratively play crucial roles in perfecting COPD outcomes, averting exacerbations, and augmenting healthcare effectiveness. By addressing modifiable risk factors like smoking, advocating preventive measures like vaccination, and managing comorbidities effectively, we can lessen the burdens COPD imposes on healthcare systems and enhance patient outcomes. Additional research is necessitated to elucidate the mechanisms underpinning these interventions’ effects and to refine their implementation in clinical practice.

## Figures and Tables

**Figure 1 microorganisms-12-01437-f001:**
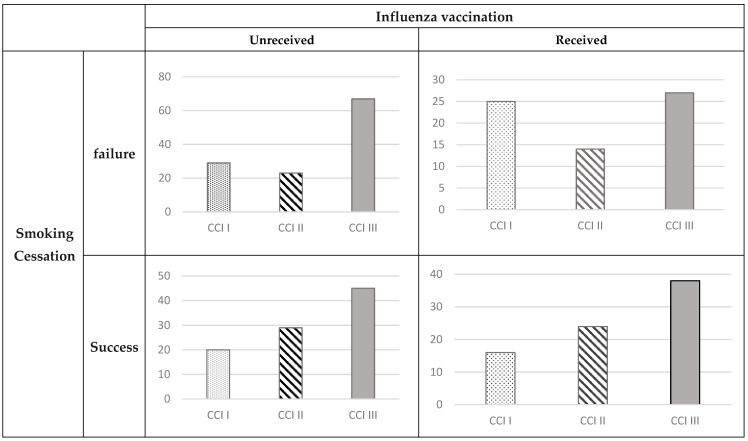
The Charlson Comorbidity Index (distribution between smoking cessation and influenza vaccination.

**Figure 2 microorganisms-12-01437-f002:**
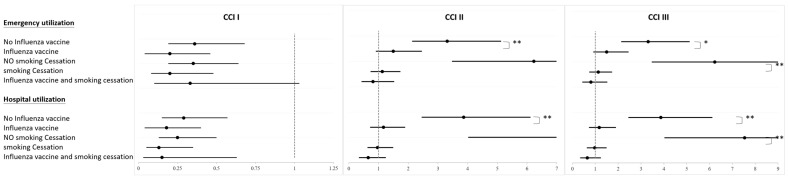
Analysis of healthcare resource utilization among influenza vaccination, smoking cessation, or both statuses in each CCI group (* *p* < 0.05, ** *p* < 0.01).

**Figure 3 microorganisms-12-01437-f003:**
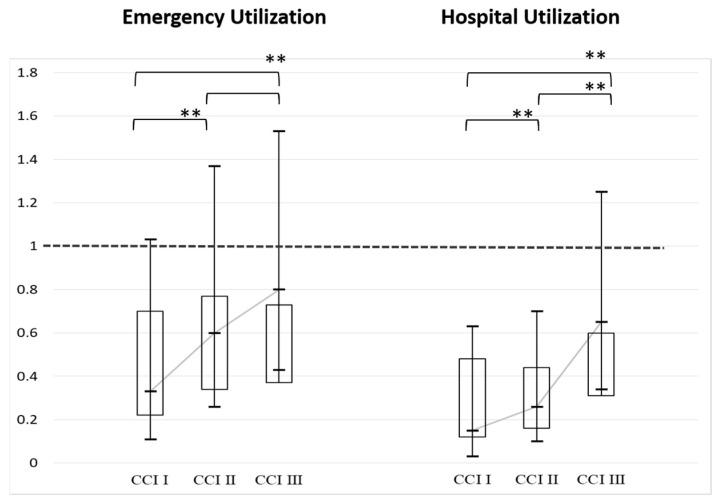
Odds ratios analysis of emergency utilization and hospital utilization for those who received the influenza vaccine and successful smoking cessation in CCI I (CCI score: 0–3), CCI II (CCI score: 4–6), and CCI III (CCI score: 7 or higher, ** *p* < 0.01).

**Table 1 microorganisms-12-01437-t001:** The demographic characteristics of enrolled 357 patients were categorized into three groups based on their Charlson Comorbidity Index (CCI) scores: CCI I group (CCI scores: 0–3), CCI II group, (CCI scores: 4–6), and CCI III group (CCI scores: 7 or higher).

Variables (N = 357)	Total	CCI I	CCI II	CCI III	*p* Value
Number		90 (25.21%)	90 (25.21%)	177 (49.58%)	
Age (SD)		71.69 (8.45)	71.03 (10.04)	70.57 (9.63)	0.233
Gender					0.067
women	27 (7.56%)	2 (2.22%)	7 (7.78%)	18 (10.17%)	
man	330 (92.44%)	88 (97.78%)	83 (92.22%)	159 (89.83%)	
Influenza vaccination					0.348
Yes	213 (59.66%)	49 (54.44%)	52 (57.78%)	112 (63.28%)	
No	144 (40.34%)	41 (45.56%)	38 (42.22%)	65 (36.72%)	
Smoking Cessation					0.036
Yes	185 (51.82%)	54 (60%)	37 (40.11%)	94 (53.11%)	
No	172 (48.18%)	36 (40%)	53 (58.89%)	83 (46.89%)	

**Table 2 microorganisms-12-01437-t002:** Multiple regression analysis of emergency visits, hospitalization frequencies, and Hospital length of stay in CCI II–III, compared to CCI I (* *p* < 0.05, ** *p* < 0.01).

**Emergency Visits**	**Coef**	**95%CI**	***p* Value**
Age	−0.02	−0.05~−0.002	0.06
Gender	0.29	−0.64~1.22	0.61
Influenza vaccine	−0.86	−1.36~−0.36	0.00 **
Smoking cessation	−1.25	−1.74~−0.76	0.00 **
CCI I (Ref.)			
CCI II	1.24	0.56~1.94	0.00 **
CCI III	1.76	1.15~2.37	0.00 **
**Hospitalization Frequencies**	**Coef**	**95%CI**	***p* Value**
Age	−0.01	−0.02~−0.001	0.06
gender	−0.68	−1.11~−0.21	0.07
Influenza vaccine	−0.59	−0.82~−0.36	0.00 **
Smoking cessation	−0.69	−0.92~−0.46	0.00 **
CCI I (Ref.)			
CCI II	0.54	0.23~0.86	0.00 **
CCI III	0.90	0.62~1.18	0.00 **
**Hospital Length of Stay**	**Coef**	**95%CI**	***p* Value**
Age	0.06	−0.07~0.19	0.9
gender	−4.39	−9.04~0.26	0.06
Influenza vaccine	−5.57	−8.07~−3.08	0.00 **
Smoking cessation	−3.15	−5.64~−0.65	0.013 *
CCI I (Ref.)			
CCI II	2.28	−1.16~5.72	0.193
CCI III	4.91	1.86~7.95	0.00 *

**Table 3 microorganisms-12-01437-t003:** Comparison of medical utilization between successful and failed smoking cessation groups and between received and did not receive influenza vaccination in each CCI group. (CCI I group, CCI scores: 0–3; CCI II group, CCI scores: 4-6; CCI III group, CCI scores: 7 or higher).

	CCI I (N = 90)			CCI II (N = 90)			CCI III (N = 117)		
**Influenza vaccine**	NO	Yes	*p* value	NO	Yes	*p* value	NO	Yes	*p* value
emergency visits mean ± SD	0.52 (1.04)	0.45 (0.91)	0.338	1.85 (2.53)	0.57 (1.15)	0.000	2.43 (3.19)	1.41 (1.86)	0.000
Hospitalization frequencies mean ± SD	0. 39 (0.80)	0.36 (0.79)	0.926	1.16 (1.25)	0.25 (0.51)	0.000	1.47 (1.67)	0.78 (1.06)	0.001
Hospital length of stay mean ± SD	3.84 (10.98)	1.60 (3.52)	0.000	6.67 (11.51)	2.22 (4.95)	0.000	10.54 (15.29)	5.48 (10.03)	0.000
**Smoking Cessation**	NO	Yes	*p* value	NO	Yes	*p* value	NO	Yes	*p* value
emergency visits mean ± SD	0.46 (0.95)	0.39 (1.05)	0.500	2.11 (3.13)	1.10 (1.54)	0.000	3.26 (3.39)	1.34 (1.90)	0.000
Hospitalization frequencies mean ± SD	0.33 (0.78)	0.20 (0.58)	0.061	1.05 (1.41)	0.60 (0.86)	0.001	1.91 (1.49)	0.78 (1.00)	0.000
Hospital length of stay mean ± SD	3.20 (10.69)	1.89 (6.79)	0.005	7.95 (14.98)	3.11 (5.49)	0.000	11.45 (13.51)	7.23 (14.36)	0.572

## Data Availability

Data can be obtained from Hui-Chuan Chang (elaine11142@cgmh.org.tw).

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
