# Peer review of "Impact of Smoking Cessation and Charlson Comorbidity Index on Influenza Vaccination Efficacy in COPD Patients"

_microorganisms, 2024, doi:10.3390/microorganisms12071437_

Round 1

Reviewer 1 Report

Comments and Suggestions for Authors

See attached file for detailed comments.

Reviewer 2 Report

Comments and Suggestions for Authors

A figure on the selection (including criteria) and follow-up of patients would be worthwhile

How was adherence to smoking cessation verified?

In improving outcomes in patients vaccinated against influenza, are there differences between the vaccines administered?

Discuss a little more about how comorbidities, including CCI, could affect the outcome of patients with COPD and how it would affect the effectiveness of influenza vaccination.

How would you apply the CCI prior to a poor patient prognosis?

Increase the size of Figure 2

Author Response

please see the attachement

Round 2

Reviewer 1 Report

Comments and Suggestions for Authors

See attached file for detailed comments.
